# Pancreatic Ductal Adenocarcinoma (PDAC): A Review of Recent Advancements Enabled by Artificial Intelligence

**DOI:** 10.3390/cancers16122240

**Published:** 2024-06-17

**Authors:** Ashwin Mukund, Muhammad Ali Afridi, Aleksandra Karolak, Margaret A. Park, Jennifer B. Permuth, Ghulam Rasool

**Affiliations:** 1Department of Machine Learning, Moffitt Cancer Center and Research Institute, 12902 USF Magnolia Drive, Tampa, FL 33612, USA; ashwin.mukund@moffitt.org (A.M.); aleks.karolak@moffitt.org (A.K.); 2School of Electrical Engineering and Computer Science (SEECS), National University of Sciences and Technology (NUST), Islamabad 44000, Pakistan; mafridi.bee20seecs@seecs.edu.pk; 3Departments of Cancer Epidemiology and Gastrointestinal Oncology, Moffitt Cancer Center and Research Institute, 12902 USF Magnolia Drive, Tampa, FL 33612, USA; margaret.park@moffitt.org (M.A.P.); jenny.permuth@moffitt.org (J.B.P.)

**Keywords:** PDAC, artificial intelligence, machine learning, screening, diagnosis, treatment, surveillance, intraductal papillary mucinous neoplasms

## Abstract

**Simple Summary:**

Pancreatic Ductal Adenocarcinoma (PDAC) remains one of the deadliest forms of cancer, characterized by high rates of metastasis, late detection, and poor prognoses. Artificial intelligence and machine learning (AI/ML) have proven to be highly effective in improving the current standard of care for many cancers, including PDAC. This review article provides a holistic overview of high-impact, transformative AI/ML applications in various areas of PDAC care. Reflecting a patient’s medical journey, these areas include screening, diagnosis, treatment, and post-treatment surveillance. Obstacles and limitations in AI/ML applications within the context of PDAC are also discussed, along with potential solutions and future directions. Collectively, this review article offers novel approaches and meaningful insights, potentially leading to solutions for the multifaceted challenges inherent in PDAC.

**Abstract:**

Pancreatic Ductal Adenocarcinoma (PDAC) remains one of the most formidable challenges in oncology, characterized by its late detection and poor prognosis. Artificial intelligence (AI) and machine learning (ML) are emerging as pivotal tools in revolutionizing PDAC care across various dimensions. Consequently, many studies have focused on using AI to improve the standard of PDAC care. This review article attempts to consolidate the literature from the past five years to identify high-impact, novel, and meaningful studies focusing on their transformative potential in PDAC management. Our analysis spans a broad spectrum of applications, including but not limited to patient risk stratification, early detection, and prediction of treatment outcomes, thereby highlighting AI’s potential role in enhancing the quality and precision of PDAC care. By categorizing the literature into discrete sections reflective of a patient’s journey from screening and diagnosis through treatment and survivorship, this review offers a comprehensive examination of AI-driven methodologies in addressing the multifaceted challenges of PDAC. Each study is summarized by explaining the dataset, ML model, evaluation metrics, and impact the study has on improving PDAC-related outcomes. We also discuss prevailing obstacles and limitations inherent in the application of AI within the PDAC context, offering insightful perspectives on potential future directions and innovations.

## 1. Introduction

Pancreatic Ductal Adenocarcinoma (PDAC) is an extremely deadly form of pancreatic cancer that accounts for over 90% of cancers in the pancreas [1]. When the exocrine duct cells that line the pancreas become cancerous, the cancer is termed Pancreatic Ductal Adenocarcinoma or PDAC. As the name suggests, the disease is of epithelial origin, describing a malignant form of pancreatic glandular tissues. There were 62,210 new cases reported in the United States in 2023, with 49,380 deaths in 2022 [2]. Of the 57,600 reported cases in 2020, 55% had already progressed to metastatic disease [3]. PDAC accounts for 2% of all cancer cases and results in 5% of all cancer deaths in the United States, thus underlining the critical need for earlier detection [4].

The mortality rates of PDAC are on the rise, increasing 1% annually, and are projected to become the second-leading cause of cancer deaths by 2030 [2,3,5,6]. Currently, pancreatic cancer is the third-leading cause of cancer death in the age groups 50–64 and 65–79 [5]. The 5-year overall survival rate for PDAC remains dismally low, under 10.8% for both metastatic and resectable cases [2]. Globally, the incidence of PDAC varies significantly, with rates exceeding 7.4 per 100,000 person-years in the United States compared to just 2.3 per 100,000 in Africa and parts of Asia. Age also plays a critical role, with mortality rates ranging from 2 deaths per 100,000 person-years among those aged 35 to 39 to 90 deaths per 100,000 for individuals over 80 [7]. Similarly, race and ethnicity are highly determinant factors in differing PDAC rates. Compared to Hispanic and Caucasian populations, African Americans have significantly greater PDAC incidence and mortality rates, as many studies present a 40 to 90% greater incidence and 10 to 20% worse survival in African American patients compared to Caucasian patients [8,9]. Variations in PDAC statistics are largely attributed to the prevalence of high-risk factors. Notably, high alcohol consumption, cigarette smoking, and type 2 diabetes have been identified as significant contributors to PDAC [10,11]. Research indicates that diabetes doubles the risk of PDAC, with 55% of patients in one study having diabetes and 25% presenting with new-onset diabetes—a frequent precursor to PDAC diagnosis [10]. Furthermore, cigarette smoking for 50 pack-years has been shown to increase the risk of PDAC by 91%, and the risk is decreased by 9% for every year quit [12].

Due to the ambiguity of the symptoms presented by pancreatic cancer patients and the low incidence of the disease, there is no standard universally accepted screening or early detection procedures for sporadically formed PDAC [13]. Guidelines exist for identifying individuals at increased risk of developing PDAC because of a familial or inherited predisposition [14]. These individuals usually fall into the following categories: having first-degree relatives with the disease, family members with early onset pancreatic cancer, family members with an inherited genetic syndrome of PDAC, family members diagnosed with PDAC in two or more generations prior [14]. According to McGuigan et al., individuals with a familial risk of pancreatic cancer are advised to begin screening at the age of 50 for suspicious and non-suspicious cysts or lesions [15]. The screening involves histopathological examination or radiological imaging via Computed Tomography (CT), Magnetic Resonance Imaging (MRI), or Endoscopic Ultrasound (EUS) [15,16].

The cysts and lesions that are screened via MRI, CT, or EUS/CE-EUS usually fall into three categories that may result in PDAC pathogenesis: Intraductal Papillary Mucinous Neoplasms (IPMNs), Pancreatic Intraepithelial Neoplasia (PanIN), and mucinous cystic neoplasms [17,18]. PanINs and IPMNs involve the ductal system, while mucinous cystic neoplasms do not [13]. PanINs are microcystic lesions <5 mm, while IPMNs and mucinous cystic neoplasms are macrocystic lesions >5 mm [13]. Importantly, precursor lesions are common, as PanINs occur in over 75% of older adults [13]. This signifies that only a small percentage of pancreatic lesions serve as precursors for PDAC. Unlike PanIN lesions, IPMNs are observable on CT/MRI/EUS imaging, and 20–30% of PDAC cases result from these lesions [19,20]. Pancreatic lesions with the potential for malignancy are characterized based on their neoplastic status, where they develop within the pancreas, and/or their cell type or physiological role [17]. For example, a branch duct IPMN is an intraductal neoplastic cyst in the branch duct of the pancreas that produces mucin [17]. The resulting characteristics are defined through the imaging modalities mentioned previously [17,18]. Standard procedures for detecting invasive carcinoma in IPMNs and other lesions include pathology-based staining, typically employing Hematoxylin and Eosin (H&E) and immunohistochemistry (IHC) [16]. After a confirmed diagnosis of PDAC, the pancreas of patients is usually screened using imaging-based methods such as EUS, CT, and MRI. [15,21]. Some studies highlight that out of these choices, EUS provides better sensitivity [15]. In contrast, others emphasize the advantage of CT, which has an average detection sensitivity of approximately 86% and can delineate the hypodense tissue characteristic of PDAC [13].

The standard approach for patients with resectable tumors is surgery followed by FOLFIRINOX chemotherapy (54.4 months overall survival) or single-agent gemcitabine treatment (35 months overall survival) [22]. For tumors that are borderline resectable, neoadjuvant systemic therapy that is either accompanied or not accompanied by radiation is considered the standard procedure [22]. One reason is the possible recurrence of distant metastases that are not completely resected [3]. The patients with locally advanced tumors are given systemic therapy followed by radiation [22]. Lastly, patients with advanced-stage tumors are given a combination of non-surgical interventions, including multiagent chemotherapy regiments, PDAC, and gemcitabine [22]. The most common chemotherapy regimens administered for PDAC are gemcitabine, FOLFORINOX, and nab-paclitaxel, and can be administered combinatorially with other forms of therapy [23]. Targeted therapies could also be administered alongside a chemotherapy agent, and the use of EGFR (Epithelial Growth Factor Receptor) inhibitors (erlotinib) in combination with gemcitabine is a focus of many clinical trials [24,25]. In patients that harbor *BRCA* (Breast Cancer Gene) mutations (about 5%), Olaparib has recently become FDA-approved as a form of maintenance therapy [3].

The use of artificial intelligence (AI) and machine learning (ML) has significantly increased in the healthcare space in recent years [26,27,28,29]. Like many other cancers, PDAC screening, diagnosis, and treatment planning can be aided by AI models and ML algorithms. Consequently, this application has been the focus of many clinical trials. Table 1 highlights the current clinical trials involving pancreatic cancer and artificial intelligence. The purpose of our study is to consolidate current literature highlighting the use of AI to aid in the screening, diagnosis, and treatment of patients with PDAC. Although other studies exist that have summarized applications of AI in pancreatic cancer, this review differs by aggregating this information according to a patient’s chronological timeline and concurrently providing a detailed overview of how each AI/ML application aims to aid in the process [30,31,32,33]. Just as a patient first receives screening, receives a confirmed diagnosis, and then is placed on a treatment plan, the review’s sections correspond to a patient’s odyssey.

The review is organized as follows. Section 2 describes the applications of AI in PDAC screening, and Section 3 describes applications in the monitoring and progression of IPMNs. Later, in Section 4 we focus on PDAC diagnosis, and in Section 5 we focus on models for PDAC treatment. In Section 6, post-treatment surveillance is addressed. Section 7 presents the key challenges related to applying AI models for PDAC screening, diagnosis, and treatment and the future directions where additional AI applications could potentially help. Finally, Section 8 concludes the review.

## 2. Application of AI/ML Models in PDAC Screening and Early Detection

Numerous studies highlight the significant challenge in the early detection of PDAC, which is frequently diagnosed at and therefore results in predominantly unresectable cases [10,30]. Compounding this issue is the lack of biomarkers outside of the *CA-19-9* biomarker, the only FDA-approved biomarker for PDAC [15,34]. Given that approximately 10% of PDAC cases stem from hereditary mutations, there is a pressing need for improving screening methods to identify the remaining 90% of patients who develop sporadic PDAC [35]. The fact that most patients remain asymptomatic until the disease progresses to an unresectable stage necessitates the utilization of known risk factors in PDAC pathogenesis as a basis for early detection of high-risk individuals [30]. For instance, with studies indicating that 25% of PDAC patients exhibit new-onset diabetes and 50% already have the disease at diagnosis, diabetes mellitus emerges as a significant risk factor and a potential clinical marker for predicting PDAC risk in the future [10]. A summary of recent studies and details of AI models, datasets, and evaluation metrics is presented in Table 2.

### 2.1. Classical ML Models for PDAC Risk Prediction

PDAC risk prediction is the first step in screening and surveillance, as a risk score above a certain threshold can alert primary care physicians, oncologists, pathologists, and radiologists to examine the pancreas. Several studies using classical ML for predicting PDAC risk were summarized by Hayashi et al. in their review article [30].

More recently, Muhammad et al. developed an Artificial Neural Network (ANN) model to stratify patients into low-, medium-, and high-risk categories for PDAC based on a range of personal health features collected from a cohort of PDAC patients before diagnosis [36]. Given the often asymptomatic nature of PDAC, this study holds considerable clinical significance, as individuals classified within the high-risk category could be prioritized for comprehensive screening programs aimed at early malignancy detection. The authors used a dataset comprising 800,114 samples from the National Health Interview Survey (NHIS) database, allocated into a 70–30% training-testing split. The model demonstrated a sensitivity and specificity of 80.7% and an Area Under the Curve (AUC) of 0.85 on the test dataset. The ANN was trained using 18 features encompassing personal health data, dietary habits, and genetic history from patients enrolled in the National Cancer Institute’s (NCI) prostate, lung, colorectal, and ovarian cancer trials in 2018. The model architecture included an input layer of 18 neurons, two hidden layers with 12 neurons each, and a single output neuron. A specific threshold was employed to categorize patients as either at risk of cancer or not. Subsequently, these categorizations were adjusted to capture low-, medium-, and high-risk, enabling the immediate initiation of screening procedures for those identified as high-risk.

Applebaum et al. developed a logistic regression model to predict the risk of PDAC using 18 clinically relevant features extracted from EHR (Electronic Health Record) data [37]. These features include pancreatitis, diabetes, jaundice, chest pain, abdominal pain, and others. Given the relatively low incidence rate of PDAC, EHR data from patients before diagnosis are both highly valuable and scarce. The inclusion of direct symptoms of PDAC, such as pancreatitis, jaundice, and abdominal pain, alongside the occurrence of diabetes in the model, underlines its clinical relevance. With an AUC of 0.71, the model demonstrates good efficacy in identifying high-risk individuals. The logistic regression model was trained using a dataset comprising 594 PDAC cases and slightly over 100,000 non-PDAC cases. Model validation was performed with 408 PDAC and 160,185 non-PDAC cases. The findings revealed that the model identified three to five times more high-risk patients than initially captured by the dataset.

### 2.2. Natural Language Processing (NLP) for PDAC Risk Prediction

Using patient’s medical records, clinical characteristics, and other text-based data archived in EHR is emerging as a significant resource for the risk prediction and early detection of PDAC [43,44]. This approach is particularly advantageous when contrasted with the complexities associated with imaging techniques for PDAC detection [38,43].

The descriptors of the word *pancreatic cyst* from the National Library of Medicine (NLM) and the Unified Medical Language System (UMLS) were used as concepts to identify patients diagnosed with cysts [38]. The authors collected a dataset of 566,233 medical reports describing pancreatic cysts to identify keywords. Later, these were manually refined by adding relevant descriptors and removing irrelevant ones [38]. The study utilized a rule-based algorithm to identify whether each sentence in each medical record contained one of the identified descriptors, which used a query-based technique. A dependency parser was also created to validate whether the descriptors were affirmed or negated based on the surrounding words using a probabilistic framework [38,43].

The use of NLP techniques applied to pathology reports may help with better risk prediction for PDAC outcomes. Pathology reports contain information about a cyst or tumor’s size, location, and diagnostic conclusions by the pathologist [39]. Recently, a study demonstrated the use of named entity recognition (NER) alongside generative ML comprehension to construct a prognosis prediction model. The study used TCGA-PDAC pathology reports—1044 for model training, 448 for internal validation, and 165 for external validation. The model used a pre-trained encoder-decoder framework and an autoregressive Transformer to identify and generate entities alongside their corresponding tag indices. These entities and sequence tags, encapsulating morphological features of the PDAC, were subsequently utilized as key indicators for the Tumor, Node, Metastasis, Other, and Resection (TNMOR) classification system. The model demonstrated robust performance, achieving precision and recall scores of 88.83% and 89.39%, respectively. Although the study used data from patients who were already diagnosed with PDAC, the entities and sequence tags generated through the model could be used in the risk prediction of patients containing suspicious cysts and lesions such as IPMNs and allow pathologists to place such patients into the proper treatment plans depending on their severity.

In another study, the International Classification of Diseases (ICD-8 and ICD-10) codes were used as features for analyzing the health records of approximately 8.6 million patients [40]. The data, drawn from the Danish National Patient Registry and the United States Veterans Affairs system, were processed using Transformer models to predict PDAC risk over various time frames [40]. The dataset was partitioned using an 80-10-10 split for the model training, validation, and testing. The model achieved an AUC of 0.88 on the test data. The study’s findings suggest the potential of the Transformer-based model for PDAC screening and early detection. Specifically, among the top 1000 patients ranked by risk, 70 additional individuals were identified as high-risk by the modes based on their EHR data [40].

### 2.3. Computer Vision (CV) for PDAC Risk Prediction

CV techniques include AI/ML models specialized for processing imaging data. The CV techniques for screening high-risk individuals for PDAC may involve various imaging modalities [43]. The evolution of PDAC is marked by numerous morphological and textural transformations, including the emergence of IPMNs, other neoplastic changes, pancreatitis, or dilation of the pancreatic duct. These transformations and early indicators can be potentially monitored with imaging techniques before a clinical diagnosis is established [21,31]. Despite CT scans being the gold standard for precancerous imaging in PDAC, physicians often face challenges in identifying these critical features unaided, as many changes are not discernible to the naked eye [31]. CV models can help recognize these subtle cues, thereby identifying high-risk individuals. Indirectly, this capability not only facilitates more informed treatment decisions but also aids in pinpointing potential biomarkers for PDAC’s onset and progression [31].

The Felix Project focuses on detecting PDAC from CT scans containing smaller tumors, which are typically difficult to detect [43]. The project used manually segmented CT images and a specialized deep neural network, U-Net, for pixel-level segmentation of objects of interest in an image [42]. First, the model was trained to recognize the normal pancreas and neighboring abdominal organs and then identify PDAC within the pancreas [42]. The model was trained on 3192 scans and tested on 1846 scans, yielding sensitivity, specificity, and Dice Score Coefficient (DSC) scores of 93%, 99%, and 0.653, respectively [42]. The study claims it has the capabilities of a *second-reader*, an additional independent reader, who can point out lesions of interest [42].

Qureshi et al. conducted a retrospective study focused on patient stratification based on risk, utilizing CT scans to compare pre-diagnostic PDAC images with post-diagnostic counterparts [41]. The study used a relatively small cohort of 108 cases but extracted a comprehensive set of 4000 radiomics features from these images. Using Recursive Feature Elimination (RFE), the features were distilled to those exhibiting significant statistical trends, with either steady increases or decreases in their values, reducing the number of features to only 4.5% of the original features. Naive Bayes classifiers with an ensemble learning framework were trained using five features at a time to classify patients into healthy or precancerous categories. This approach underscores the potential of targeted feature selection in enhancing the accuracy and efficacy of PDAC risk stratification models.

## 3. Application of AI/ML Models in the Monitoring of IPMNs

IPMNs and other cysts that grow abnormally on the pancreas can potentially become malignant. It is estimated that 20 to 30% of IPMNs develop into PDAC [20]. As a result, radiological imaging and biopsies are performed to monitor the progression of these lesions [21]. The diagnosis and subsequent management of such neoplasms are guided by the Fukuoka guidelines, which set forth specific criteria for identifying lesions that necessitate intervention [45]. For the lesions that meet these criteria, the recommended standard of care is adapted to prevent the potential transition to PDAC, which may involve surgical resection, such as a Whipple’s procedure or total/distal pancreatectomy [45].

Identifying high-grade dysplasia or invasive carcinoma in IPMNs poses significant challenges owing to the detection accuracy of conventional methods such as EUS and cyst fluid analysis [46]. Given that most IPMNs are benign or exhibit low-grade dysplasia, thereby negating the need for surgical resection, CV models present a more efficient diagnostic alternative to traditional approaches that rely on confocal microscopy. A summary of recent AI studies focused on differentiating between high- and low-grade IPMNs and relevant details about models, datasets, and evaluation metrics is presented in Table 3.

Recently, Jiang et al. proposed a dual CNN model to increase the accuracy of the PDAC diagnosis process [46]. The first CNN (Convolutional Neural Network), a VGGNet, processes video data for risk stratification. The second CNN, a Mask R-CNN, identifies papillary structures from video frames and extracts critical radiomic features, such as epithelial thickness. These features are particularly effective in distinguishing between invasive and non-invasive IPMNs. The first model achieved an accuracy of 85.7%, and the second model reached an accuracy of 82.9%, both surpassing the performance of the current standards of care in utilizing confocal microscopy data for diagnosis, which stand at 74.3% and 68.6% for risk stratification and papillary structure segmentation, respectively. To further enhance diagnostic precision, the authors proposed a multimodal model that integrates various data types, including clinical features, cyst morphological characteristics, cyst fluid analysis, and biopsy data from confocal endomicroscopy.

To address the challenge of unnecessary surgeries for low-grade IPMNs, Watson et al. used CNNs to evaluate resected pancreatic cystic neoplasms. Using axial CT scans of the neoplasms, the authors trained a LeNet model comprising three convolutional layers, a flattening layer, and two fully connected layers to predict the grade of cystic lesions [45]. The authors aimed to mitigate the limitations and inconsistencies associated with the Fukuoka guidelines, which sometimes lead to unnecessary resections of benign lesions or missed opportunities for resecting advanced-grade IPMNs. The study underscored the problematic nature of relying solely on the Fukuoka guidelines, noting that nearly 20% of benign lesions undergo unnecessary resection. Given the necessity of a biopsy for a definitive diagnosis, many patients undergo needless procedures. Watson et al.’s model demonstrated remarkable efficacy, accurately determining the grade of cystic lesions in eight out of nine cases within the testing dataset, resulting in an accuracy rate of 89%. This significantly outperformed the Fukuoka guidelines, which correctly classified only six out of nine lesions. The authors highlight the model’s potential to prevent unnecessary surgical procedures and identify advanced-grade neoplasm, offering a more reliable and efficient diagnostic tool than the existing guidelines.

A high morbidity rate is associated with IPMN surgeries, approximately 40%, which necessitates the need to correctly identify cysts that do not require surgical resection, many of which are low-grade dysplasia [47]. Recently, Hernandez-Barco et al. trained a Support Vector Machine (SVM) to predict the grade of dysplasia in IPMNs. They integrated clinical data, such as jaundice, pancreatitis, diabetes, and abdominal pain, with pathological features derived from resected IPMNs, including the presence of nodules, septations, and pancreatic duct dilation. The study included data from 575 patients, with a 4:1 ratio for the training and testing datasets. The model demonstrated an accuracy of 77.4% for distinguishing between low- and high-grade IPMNs.

## 4. Application of AI/ML Models in PDAC Detection and Diagnosis

Detection and diagnosis both refer to discovering the presence of cancer. *Detection* refers to distinguishing healthy individuals from those with cancer, primarily through imaging modalities such as CT or EUS [32]. *Diagnosis*, on the other hand, involves the specific identification of PDAC as opposed to other types of pancreatic lesions, typically verified through histopathological examination. The imaging techniques can contribute to this process as well [32]. CT scans, with an average sensitivity of approximately 86%, are widely used for detecting PDAC, where the cancer usually appears hypodense. Despite Contrast-Enhanced CT (CECT) being the most common detection method, other techniques such as MRI, Magnetic Resonance Cholangiopancreatography (MRCP), and EUS are also employed [15,46].

Despite being a time-consuming and labor-intensive process, the evaluation of Whole Slide Images (WSIs) by pathologists is currently the gold standard for diagnosing PDAC [48]. This process is further compounded by the limited accuracy of CT imaging in detecting PDAC before the onset of metastases, which is estimated to be around 50% [43]. The difficulty in detecting PDAC at this early stage is partly due to the significant microscopic variations in textural and morphological changes within the pancreas. AI models and ML algorithms have the potential to identify subtle changes that are not easily detected through traditional imaging methods, thereby enhancing a physician’s ability to detect PDAC at earlier stages [31]. Integrating AI into the diagnostic workflow offers an opportunity to reduce the time needed for data processing and decision-making. This will streamline the process of identifying PDAC and facilitate earlier intervention, potentially improving patient outcomes [49].

A summary of a set of recent AI studies focused on the detection and diagnosis of PDAC, along with details about the models, datasets, and evaluation metrics, is presented in Table 4.

### 4.1. PDAC Detection

In a recent study, Mukherjee et al. used radiomics features to detect PDAC at the pre-diagnostic stage, approximately 3–36 months before the confirmed clinical diagnosis [50]. The study extracted a total of 88 radiomics features and used four different ML classification models: K-Nearest Neighbor (KNN), SVM, Random Forest (RF), and Extreme Gradient Boosting (XGBoost) [50]. The authors trained the models using 292 CT scans, of which 110 cases were pre-diagnostic and 182 were control. There were 128 CT scans in the test dataset, with 45 pre-diagnostic and 83 control cases. The SVM model showed the best prediction accuracy (92.2%), as compared to the other models and the human experts (R4 and R5 radiologists) [50]. The study highlighted the importance of radiomics features for early PDAC detection, even before a pancreatic mass is developed.

Tong et al. developed an end-to-end Deep Learning Radiomics (DLR) model for diagnosing PDAC and Chronic Pancreatitis (CP) using Contrast-Enhanced Ultrasound (CEUS) images [51]. The DLR model used multiple CNNs, ResNet-50 [62], Inception-v3 [63], VGG-16 [64], and DenseNet-121 [65], to extract rich features for image classification. A total of 558 patients with pancreatic lesions participated in the study, of which 351 patients were split into the training cohort, 109 patients into the internal validation cohort, and 98 into two external validation cohorts (cohort 1 and cohort 2). Detailed clinicopathological data of each patient was collected from three hospitals in China. The DLR model achieved an AUC of 0.978, 0.967, and 0.953 in the three validation cohorts. ResNet-50 emerged as the dominant backbone model amongst all those tested. The study indicates that their DLR model can effectively support radiologists in diagnosing PDAC and CP from CEUS [51].

Cao et al. introduced a novel architecture called Pancreatic Cancer Detection with Artificial Intelligence (PANDA) tailored explicitly for identifying and characterizing pancreatic lesions through analysis of non-contrast CT images [52]. Non-contrast CT scans of 3208 patients from a single center were used for training, while a dataset of 6239 patients from 10 distinct medical facilities was used for validation. PANDA achieved an AUC of 0.986 to 0.996 for lesion detection on the validation dataset [52]. The study’s findings suggest a promising potential for PANDA to serve as an effective PDAC detection tool, particularly within high-risk populations, given its reduced radiation exposure compared to CECT scans.

Recently, Chen et al. developed 3D TransUNet for image segmentation, leveraging nnU-Net architecture and Transformer-based encoder-decoder blocks [66]. The authors collected 2930 CT scans from a high-volume US hospital to make a large-scale pancreatic mass dataset with PDAC, Cyst, and normal pancreas labels. Of the 2930 CT scans, 1523 were identified as PDAC. The decoder-only variant of the TransUNet framework achieved a DSC of 0.626 on PDAC cases, with 89.94% sensitivity and 97.33% specificity. The encoder-only variant achieved 91.71% sensitivity on the PDAC dataset [66].

Kuwahara et al. used a CNN, EfficientNetV2-L, to distinguish between several types of pancreatic masses [54]. The authors extracted lesion images from still frames of training and validation cohorts derived from EUS videos. To address the issue of class imbalance, they generated additional images using a deep convolutional generative adversarial network (DCGAN) [67] specifically targeting the classes Autoimmune Pancreatitis (AIP), Neuroendocrine Tumor (NET), and CP. The authors generated 22,000 training images from 933 patients. The sensitivity score was 0.94 (CI: 0.88–0.98) for the diagnosis of pancreatic carcinomas, 0.96 (CI: 0.90–0.99) for PDAC, 1.00 (CI: 0.22–1.00) for acinar cell carcinoma (ACC), 0.93 (CI: 0.66–1.00) for NET, 0.73 (CI: 0.39–0.94) for AIP [54].

Viviers et al. proposed a method based on the U-Net architecture to segment the pancreatic tumors [55]. The authors collected CECT images from 97 control and 99 PDAC cases. The model utilizes external secondary features like the pancreatic duct, common bile duct, and the pancreas map, with the CT scan to aid detection. The authors reported a sensitivity of 99% on the test medical decathlon dataset, along with a DSC of 0.31 ± 0.05% [54]. The model exploits external secondary features to improve its PDAC detection performance compared to vanilla nnU-Net (sensitivity score of 92 ± 2%).

Tayebi et al. proved that differential privacy (DP) [68] training of diagnostic deep learning models is possible with excellent diagnostic accuracy, with the task of classifying the presence of PDAC [56]. In this study, 1625 3D abdominal CT images were manually labeled by experienced radiologists, and out of 1625, 867 cases were PDAC. ResNet9 architecture was used for the DP training, observing an average AUC score of 95.58% [56]. The authors have addressed the issues indicated by prior studies on using DP, causing adverse effects on model performance [69]. In this study, the calculation of privacy-fairness trade-offs, measured as Pearson’s for the PDAC dataset and on the UKA-CXR dataset (N = 193,311), vindicated the usefulness of DP in detection.

### 4.2. PDAC Diagnosis

Qiu et al. developed SVM models for CT-based texture analysis to perform tumor grading [57]. The authors acquired CECT images from 56 patients with PDAC. For classification purposes, differentiated/grade I and moderately differentiated/grade II cases were grouped and labeled as low-grade PDAC, and poorly differentiated/grade III cases were labeled as high-grade PDAC. The authors developed four models: all texture features, histogram features, run-length features, and co-occurrence features. With the all texture features model, the SVM accurately predicted low-grade/high-grade PDAC with 95% specificity, 86% accuracy, and 78% sensitivity [57].

Bakasa et al. proposed a hybrid model to extract features from CT images and classify PDAC [58]. The method automates the diagnosis and classification phases of PDAC with the same accuracy as that of an expert. After feature extraction using a CNN, classification was performed using SVMs, RFs (Random Forest), and XGBoost [70]. XGboost demonstrated superior performance over RFs and SVMs [58]. The analysis of different feature extractors, including Light Gradient Boosting Machine (LGBM), VGG16, and Inception-V3, revealed that the combination of VGG16 with XGBoost yielded 97% accuracy in classifying PDAC stages [58]. The findings of this study offer advantages in diagnosing PDAC from CT scans of the pancreas, classifying them based on the TNM staging system into five distinct labels corresponding to tumor size and metastasis status, denoted as T0, T1, T2, T3, and T4.

Cen et al. used pre-operative clinical-radiomics nomograms to differentiate between high-grade and low-grade PDAC (i.e., predict the histological grade) and predict overall survival (OS) [59]. Radiomics features were extracted from 284 CECT scans, with 200 for training and 84 for testing. An external cohort of 42 patients was used to validate the model. The model produced an AUC of 0.75 (95% CI: 0.64, 0.85) in the test cohort and 0.76 (95% CI: 0.60, 0.91) in the validation cohort [59].

Si et al. introduced a fully end-to-end deep-learning (FEE-DL) model for diagnosing pancreatic tumors from abdominal CT scans [60]. The model undergoes four steps: image screening, pancreas location, pancreas segmentation, and tumor diagnosis. The authors used a CNN, ResNet34, for PDAC classification and U-Net32 for segmentation. The model was trained on 143,945 CECT images from 319 patients and tested on 107,036 independent CT images from 347 patients. FEE-DL achieved an AUC of 0.871 and an F1 score of 88.5% on the test set. Furthermore, the model achieved 100% accuracy in identifying IPMNs and 87.6% accuracy in identifying PDAC independently [60]. The model’s average inference time per patient was 18.6 s, significantly lower than that of human experts, which takes an average of eight minutes per patient.

Ghoshal et al. proposed a Bayesian CNN for automated pancreatic cancer grading and uncertainty estimation using histopathology images, including May–Grunwald–Giemsa stain (MGG) and H&E stained images [61]. The authors curated a PDAC grading dataset comprising 138 high-resolution tissue samples stained with MGG and H&E and annotated with Normal, Grade-I, Grade-II, and Grade-III categories [71]. The dataset was divided into training, validation, and test sets in a ratio of 60% to 20% to 20%, respectively. According to the results, Bayesian DenseNet-201 model-based inference surpassed the detection accuracy of the ResNet-152V2 and VGG-19 model on their sample dataset, achieving 85.60% accuracy on PDAC grading [61]. ResNet-152V2 and VGG-19, on the other hand, achieved 83.60% and 76.52% accuracy. After obtaining predictions for all test images and sorting them by their associated predictive uncertainty, this study showed that the estimated uncertainty in model prediction strongly correlated with the classification accuracy. Thus, the estimated uncertainty was well-calibrated and could be used to avoid misdiagnosing uncertain cases and noisy datasets.

## 5. Application of AI/ML Models in PDAC Treatment Outcome Prediction and Patient Stratification

Patient survival rates for PDAC are highest when the tumor is resectable following neoadjuvant therapy [72]. However, most cases are often unresectable or have major vascular involvement and metastasis. Despite this challenge, the prediction of treatment outcomes for resectable pancreatic cancer cases is extremely important, as unnecessary surgery can often be more damaging than beneficial [73]. Similarly, resectable PDAC cases are heterogeneous, necessitating individual treatment plans [72,74]. Many studies have attempted to predict treatment outcomes or the risk for such cases using ML models [32]. A summary of recent studies and details of ML models, datasets, and evaluation metrics is presented in Table 5.

### 5.1. Patient Stratification for P-Net and PDAC Treatment Regimens

The complexity and heterogeneity of PDAC require a holistic picture of the patient’s imaging, clinical, and histopathological characteristics to determine a treatment plan properly [32,80]. There is a shortage of studies considering AI applications predicting treatment outcomes for better patient stratification in PDAC, but relevant studies exist for P-net, or pancreatic neuroendocrine tumors. The rationale, methodology, and promising results show that these studies provide a way for ML models to be used on PDAC data to better aid patient stratification.

An example of such a study considers post-treatment complications using pre-treatment images in the prediction of pancreatic neuroendocrine tumor (P-Net) grade [81]. P-Net and PDAC grading can greatly influence a patient’s treatment plan [81]. Gao et al. trained a CNN using pre-operative MRI scans and synthetic data generated using generative adversarial networks (GANs) to predict P-Net grade [81]. The study showed that augmenting the validation dataset with synthetic data has promising applications, resulting in an accuracy of 85.13% in grading rare P-Net tumors. In low-incidence tumors, such as PDAC and P-net, the challenges of small sample size can be addressed with AI models like GANs and diffusion models [81,82].

The pre-treatment image data can also be used for metastasis prediction. Metastasis occurs when the tumor is spread to other organs, and is an extremely important factor in treatment planning [83]. Klimov et al. used a dual CNN model to extend the P-Net grade classification task to risk prediction of metastasis. First, a CNN (GoogLeNet Inception V1) was trained to discriminate cancerous regions from other pancreatic tissue using manually annotated histopathology images. The model showed a discrimination accuracy of greater than 0.95%. The aggregated soft-max probability scores from the discrimination CNN were combined with additional variables (e.g., metastatic status) to calculate a metastatic score with the help of a second CNN. The second CNN (GoogLeNet Inception V1) produced an F-1 score of 0.82 in a 5-fold cross-validation test performed with 104 WSIs from metastatic patients. The model identified 13 high-risk patients, validating the model’s capability in stratifying patients for correct treatment plans. Although this study used histopathological examination on P-net tumors, it highlights the capability of annotated pre-operative WSIs to calculate metastatic risk for better patient treatment stratification in PDAC.

### 5.2. PDAC Treatment Outcome Prediction

For resectable PDAC tumors, the post-surgical complications pose an important challenge [76]. One such post-operative complication, Post-Operative Pancreatic Fistula (POPF), is a major cause of pancreatic-related morbidity due to a leakage of pancreatic fluid [30]. Since POPF can be imaged via CT scan within 4 weeks of surgical resection, using AI paired with the CT scanned images can help detect its presence prior to treatment [76]. Mu et al. used 513 pre-operative CT scans (with a 70-30 training-testing split) to train a CNN model that predicted the average probability of POPF using a gradient-weighted class activation map [76]. The model produced an AUC of 0.89 on the test dataset. The localization maps and several other clinical factors were used for downstream analyses, including multivariate linear regression to divide patients into risk groups. The model performed better than the fistula risk score (FRS), the current standard predictor for POPF. FRS uses four parameters (soft glandular texture, small-sized main pancreatic duct, undue intraoperative blood loss, and high-risk pathology) to assign the patients a score ranging from 0 to 10 [76]. An accurate prediction of high-risk individuals that may develop POPF can be used to optimize the treatment, i.e., surgical resection may be too risky.

The standard of care for resectable and non-resectable PDAC cases is chemotherapy [72]. Resectable tumors can have chemotherapy administered through adjuvant or neoadjuvant means [22]. The evaluation of chemotherapeutic response is based on tumor regression grades, where grades 0 to 2 reveal a pathological response to the therapy [77]. The prediction of such grades allows physicians to determine whether it is worth administering chemotherapy [77]. Similarly, neoadjuvant therapy is effective in increasing overall survival, but it is still important to predict the effect of the therapy [77]. Using 776 axial images from pre-operative CT scans and CA-19-9 biomarker levels, Watson et al. predicted neoadjuvant therapy response using a five-layer CNN and LeNet model [77]. The hybrid model had an AUC of 0.785, thus highlighting the strong capability of combining multimodal data to predict neoadjuvant treatment response to chemotherapy regimens [77]. Since the models use pre-operative CT scans, physicians may use such prediction to guide treatment decisions [77].

Standard chemotherapy regimens for PDAC often come with differences in drug efficacy, side effects, and long-term survival for a particular patient [22]. Therefore, it is often important for physicians to tailor a chemotherapy regimen with the best survival outcomes and the least adverse effects. The chemotherapeutic response can be predictive with significant power using ML, specifically gradient-boosted trees [84]. Wei et al. extracted transcriptomic features from PDAC patients (along with colon, breast, sarcoma, and bladder cancer totaling 2606 tumors) through autoencoders and used these features as input to a gradient-boosted decision tree to predict the response to chemotherapy [78,84]. The authors reported an AUC of 0.74 for the 5-fold cross-validation [78,84]. Similarly, Kaissis et al. used a gradient-boosted tree to distinguish between two genomic subtypes of PDAC, KRT81 positive or negative, and measured each type’s relationship to different chemotherapy regimens [79]. A gradient-boosted tree was trained to predict the overall survival for patients surgically resected, who presented KRT81 positive or negative and received a particular chemotherapy regimen. With a sensitivity of 0.92, the study found that KRT81-positive subtypes were better suited for gemcitabine therapy than Folfirinox [79].

The administration of chemotherapy regimens in PDAC patients results in major histological changes [73]. Given these changes, the identification of post-treatment histological data can reveal biomarkers that, when retrospectively analyzed, can help tailor more precise treatment regimens, avoid unwanted surgery, and be used for early detection of the disease. Additionally, the biomarkers derived from post-treatment histological data can be used to predict disease-specific overall survival. Nimgoankar et al. explored this relationship by using cellular segmentation through the Hover-Net model to extract histology signature profiles of cells following adjuvant gemcitabine treatment in 47 PDAC patients. Extracted features from histopathological data were then used to create a single histological signature that was statistically analyzed to show significance in disease-specific survival metrics. The results showed a hazard ratio of 2.94, indicating that the histological signature was strongly associated with disease-specific survival. By deriving such biomarkers, Nimgoankar et al. show that the histological signatures can be used for survival prediction and help physicians decide whether a patient is suitable for neoadjuvant gemcitabine therapy.

## 6. Application of AI/ML Models in Patient Surveillance of Post-Treatment Complications from PDAC Treatment

The strategies developed for detecting and diagnosing PDAC can be used to monitor post-treatment effects in patients who have received surgical resection of tumors [32]. This allows for identifying micro-metastases or deep lymph node involvement [32]. Similarly, unresectable cases, which comprise the vast majority of PDAC patients, undergo chemotherapy and are examined using CT scans to derive morphological features for further assessment. Physicians routinely use these data to evaluate the treatment quantitatively. In treatment where tumors are resected, histopathological quantification is used to derive tumor regression grade (TRG), a measure of the amount of residual cancer following tumor destruction. In treatments without resecting tumors, morphological tumor features are used to calculate a quantitative chemotherapeutic performance metric known as response evaluation criteria in solid tumors (RECIST). However, both post-treatment evaluation metrics suffer from limitations stemming from tumor heterogeneity and lack of standardization. Modern AI and ML models may address these limitations by using the radiological and histopathological imaging data gathered through monitoring treatment outcomes. Integrating this data with finely tuned models allows the generation of more precise, personalized, and standardized post-treatment metrics. This section presents recent studies and details of ML models, datasets, and evaluation metrics, which are also summarised in Table 6.

Examining the size and shape of remaining metastases is a crucial aspect of post-treatment evaluation of resectable tumors [85]. Pan et al. proposed to automate this examination process using an ML model that quantifies spatial features of metastases in mice after receiving therapeutic monoclonal antibody (for carbonic anhydrase XII) treatment for pancreatic, breast, and lung cancer. Amplified 3-D full-body fluorescence light microscopy was performed to collect 3D image data from the antibody-treated mice. The data was fed into a 3-D U-Net model to output a 2-D map of every pixel’s probability of being metastatic. With an F-1 score of 80%, the model accounted for 29% increased detection of micro-metastases compared to a physician’s manual identification. The authors state that, through this model, the detection of micro-metastases can reveal which regions are targeted by the clonal antibody therapy. Similarly, detecting micro-metastases across the entire body provides the data required to elucidate why certain regions are eliminated by antibody therapy, and others are not.

Post-treatment surveillance data in resected pancreatic lesions can guide personalized treatment of PDAC patients with early-stage disease [32]. Typically, chemotherapy regimens are administered through adjuvant or neoadjuvant means based on the physician’s decision, and the combination of resection plus chemotherapy yields promising results [22]. Of these chemotherapy regimens, mFOLFIRINOX is considered the state-of-the-art, showing the best overall survival outcomes, but gemcitabine and gemcitabine plus capecitabine are also commonly used [87]. However, it is seen that different chemotherapy regimens affect the morphological nuclear characteristics of the resected tumors in neoadjuvant therapy differently. Krishna et al. explored this relationship by calculating the correlation between morphological nuclear biomarkers of a resected PDAC tumor and the therapy that it received [87]. To derive morphological nuclear tumor features, the authors developed a segmentation model on 139 histopathology slides of resected pancreatic tumors receiving gemcitabine or FOLFIRINOX. Although the exact architecture of the model was not specified, the segmentation model extracted nuclear features of the histopathology slides and correlated them against disease-specific survival. The concordance index (C-index) was measured for correlations between each nuclear morphological feature against disease-specific survival, where the tumor was treated with either gemcitabine or FOLFIRINOX treatment [87]. The promising results showed that elliptical features in the nuclei geometry had a strong correlation with gemcitabine treatment. No statistical correlation linked 5-FU patients’ geometric nuclei features to disease-specific survival, indicating the model was more appropriate for gemcitabine-treated patients.

The evaluation of the effectiveness of neoadjuvant chemotherapy for treating PDAC tumors can also be performed through the tumor response scoring system (TRS), a measure of residual tumor burden [86]. Following surgical resection, histopathology examination of the remaining pancreas can be performed [86]. These slides are examined and scored based on TRS but often lack standardization [86]. Janssen et al. propose a standardized approach using ML models to segment such histopathological images [86]. By using segmentation masks over the tumor, epithelium, and normal ducts for labels, several U-Net models were trained, tested, and validated [86]. The authors used 50 images for model training, 5 for validation, and 9 for testing. Through testing via U-Net with several different types of encoders (DenseNet161, DenseNet201, EfficientNet, ResNet152) on the same histopathological data, results showed that DenseNet161 had the highest performance, showing an F-1 score of approximately 86% [86]. The superior performance of DenseNet161 allows for the possibility of automatic segmentation of the pancreatic ducts along with the residual tumor in heterogeneous cases of PDAC [86]. Through the segmentation of important pancreatic components, this data can be used as a standardized approach that replaces TRS and inter-observer variability is effectively eliminated. Moreover, Jannsen et al. explain that the segmentation predictions can identify key biomarkers related to a variety of clinical outcomes [86].

## 7. Discussion

### 7.1. Challenges and Opportunities

The studies reviewed in this article offer an optimistic perspective on the applications of AI across the entire spectrum of PDAC, from initial screening and detection to treatment and surveillance. Please refer to Figure 1 and Figure 2 for a graphical representation of some of the key ways in which AI has impacted PDAC care. Figure 1 provides a high-level overview of the different areas of impact of AI in PDAC, whereas Figure 2 shows the AI applications that intersect different aspects of patient care for PDAC. This section identifies the principal limitations of current AI models and the obstacles to their clinical implementation.

Diagnosing PDAC typically requires an analysis of histopathology slides, various forms of radiological imaging such as EUS, CT, or MRI, along with molecular testing, routine laboratory work, and other relevant clinical information [15,30]. The diversity of data types and their sources, the specific data processing techniques required for each modality, and the subjective interpretation of these data by different medical specialists all contribute to the challenge of creating a universally applicable AI framework for autonomous PDAC diagnosis. Researchers often concentrate on isolated sub-problems, gathering and annotating data to develop ML models tailored to these specific issues. Yet, this approach might not align with the clinical reality, where data collection and analysis are iterative (e.g., additional lab tests, imaging, or biopsies may be required), and decision-making often involves several rounds of consultation among specialists. There is a clear need for the development of human-centric, multimodal, hierarchical AI systems capable of processing various data types, requesting additional information when necessary, and incorporating the insights of human experts into the final diagnosis.

We have presented various use cases where CV and NLP techniques enhance early detection, screening, diagnosis, and treatment planning for PDAC. However, a significant obstacle in developing highly accurate models is the necessity for a large and diverse dataset. Currently, gathering a sufficiently large and varied dataset for PDAC images is deemed unfeasible for the general population. The rare occurrence of PDAC, combined with the challenges in predicting the malignant potential of pancreatic cysts, results in the infrequent imaging of precancerous lesions, complicating the task of risk prediction [13,31]. Moreover, the low incidence of PDAC, alongside its high metastatic rates, presents a persistent issue of class imbalance, posing difficulties for machine learning models in screening, diagnosis, treatment, and surveillance efforts [30]. To overcome these challenges, collaborative efforts are needed to develop centralized, anonymized PDAC imaging and clinical databases, facilitating the creation of more diverse and extensive datasets that can enhance AI model training and validation.

The inherent heterogeneity of PDAC tumors, manifesting in their varied shapes, sizes, locations, and especially their progression rates, significantly complicates the development of accurate AI models. This diversity necessitates that AI solutions not only recognize but also accurately classify the multitude of PDAC subtypes along with their distinct morphological features and growth trajectories. For AI models to be clinically actionable and relevant across different healthcare settings, they must be trained on data that reflect this broad spectrum of PDAC characteristics. Such comprehensive training enables the models to develop a generalized understanding of the disease, ensuring their applicability and utility in diverse clinical environments [13,32]. To address this challenge, it is imperative to adopt a multifaceted approach involving the aggregation of large, diverse datasets and interdisciplinary collaboration among oncologists, pathologists, and AI researchers to enrich AI training environments with a wide array of PDAC presentations. Federated learning techniques can be used to train AI models without sharing the data [88].

Current research on predicting treatment outcomes for PDAC often centers on the efficacy of adjuvant or neoadjuvant therapies in cases where the tumor is resectable, despite the fact that the vast majority of PDAC cases are unresectable at diagnosis [30,72]. There is a pressing need for studies to focus on treatment alternatives specifically designed for unresectable PDAC cases. Such research should aim to identify effective therapeutic options and establish critical time points at which a particular chemotherapy regimen may cease to benefit the patient, thus optimizing patient care and resource allocation [32].

Interpretability presents a persistent challenge in applying AI to cancer diagnostics and treatment, as the complex features and representations derived from AI models, like those from multiple convolutional layers, are often not intuitively understandable to humans. These complex model components are crucial for the ML model’s semantic understanding and ability to make highly accurate predictions. Yet, clinicians, who often seek to understand the causal relationships and underlying biological mechanisms of disease progression, find the abstract weights, features, and embedding methods used in AI models to be opaque and not directly correlated with the known biological aspects of a disease [30,32]. This gap underscores the need to bridge AI model outputs with clinically relevant insights, enhancing the models’ utility in practical healthcare settings.

### 7.2. Future of PDAC Research: Pancreatic Cyst Monitoring

AI applications have the potential to significantly enhance the current PDAC care standards, with the most critical area being the early detection of PDAC through the monitoring of suspicious cysts and lesions. The urgency for early detection stems from several compelling observations. Firstly, an estimated 55% of PDAC cases have already metastasized by the time of diagnosis, often making it too late to tailor a personalized treatment plan [2]. Once metastasis occurs, the option for surgical resection becomes impractical, which severely impacts survival rates. Secondly, research indicates that precursor lesions are a common occurrence with age, as they are present in over 75% of older adults [13]. Given the potential for these lesions to turn malignant, there’s a pronounced need for early detection efforts to be expanded globally [13].

Enhanced screening efforts to monitor early/cancerous pancreatic lesions may not only facilitate the identification of more patients with malignancies but also generate a larger and more diverse dataset for AI models to utilize. The classification of pancreatic cysts and lesions by AI has already demonstrated high accuracy and clinical relevance in diagnosis and detection, suggesting the potential of AI models to effectively track the progression of IPMNs from pre-diagnostic images [61,71]. Adopting AI can address the inconsistencies and manual labor intensiveness inherent in the Fukuoka guidelines and pathology, enhancing the standardization of diagnostic criteria. This improvement in early-stage pancreatic cancer detection is expected to increase resection rates, improve survival outcomes, and provide an enriched dataset for AI models dedicated to PDAC diagnosis, treatment, and post-treatment monitoring.

### 7.3. Future of PDAC Research: Incorporation with Molecular Data

Despite progress in PDAC classification by AI, challenges persist in diagnoses, stratification, and treatment due to factors such as lack of distinctive clinical symptoms or specific molecular markers, and high tumor heterogeneity. As discussed in this review, AI enhances and highlights the need for further improvement in early-stage detection and various levels of cancer development. Given these challenges, the integration of imaging with molecular data emerges as one of the next steps to enrich datasets, analyses, and improve survival outcomes. To date, such studies are challenging and rather rare, highlighting the limitations of select data sets to extract PDAC molecular signatures or with most focusing on multiple cancer types [89,90].

Within molecular modalities, approaches combining various molecular data types, such as DNA, RNA, or protein-related data, are common. These data types often include transcriptomics, proteomics, metabolomics, or epigenetic modifications like DNA methylation. While most ’biomarker-detecting’ models have addressed one data type, more multi-omics analyses are emerging [84]. For instance, Osipov et al. proposed the Molecular Twin AI platform using clinical and multi-omics data to predict PDAC patients’ outcomes [34]. Another example by Sinkala et al. involves subtyping pancreatic cancer cell lines using multiple biomarkers, including mutation, methylation, protein expression, and miRNA, leading to the identification of two clinically distinct subtypes [91]. Epigenetic signatures, particularly DNA methylation, were significantly different between these subtypes, highlighting the importance of studying genetic material beyond DNA sequence.

Additionally, the understudied epigenetic modifications, including histone modifications, were highlighted as potential biomarkers for PDAC by Elrakaybi et al. [92]. However, investigations of biomarkers and treatments focusing on epigenetic modifications still require integration with clinical data to identify patients who will benefit from a given treatment. The utilization of epigenetics also holds promise in PDAC diagnostics and treatment response, especially when combined with high-resolution imaging data, as epigenetic marks often disrupt DNA organization in the nucleus. As proposed by Bahado-Singh et al., circulating cell-free DNA can be a minimally invasive source of biomarkers for PDAC detection by utilizing epigenetic data, such as DNA methylation [93].

Furthermore, additional approaches beyond traditional biomarker definitions, such as the microbiome, are gaining attention. For example, Li et al. proposed the ML Random Forest model differentiated metastatic vs. non-metastatic PDAC among patient samples, with microbial markers showing significant differences compared to other molecular analyses [94].

Beyond biomarker discovery, the applications of ML/AI in PDAC extend to early detection and treatment optimization. Predictive ability of molecular profiles on response to gemcitabine was evaluated with multiple ML algorithms by Ogunleye et al. [95]. However, challenges with data quality and availability persist. While most reports up to date use statistical methods and platforms, the AI applications to predictive PDAC precursor evaluation in pancreatic cysts and tumors are emerging [96,97,98,99]. Cross-disciplinary collaborations can advance data sharing and model interpretability, making continued studies on complete multi-omics data integration and incorporation with imaging data essential focuses of current research in PDAC.

## 8. Conclusions

In recent years, AI has emerged as a pivotal tool in enhancing the screening, detection, diagnosis, treatment, and surveillance of PDAC. AI can potentially reduce the workload of physicians, transform cancer detection methodologies, and increase prognostic accuracy. Despite these advancements, the field still requires extensive validation through prospective multicenter studies and larger PDAC datasets. Current research on AI-assisted histopathologic diagnosis of PDAC is sparse, and the variability in histopathology staining and imaging techniques presents substantial hurdles in developing universally applicable models. Nonetheless, this review article offers an extensive overview of AI’s role in elevating the standard of care for PDAC, covering aspects from screening to post-treatment monitoring. With the increasing availability of diverse data, including clinical notes, biometrics, imaging, and histopathology slides, there is a significant opportunity for researchers to develop more precise and broadly applicable AI models. These efforts aim to enhance the quality of life for PDAC patients, offering hope in the fight against this formidable disease.

## Figures and Tables

**Figure 1 cancers-16-02240-f001:**
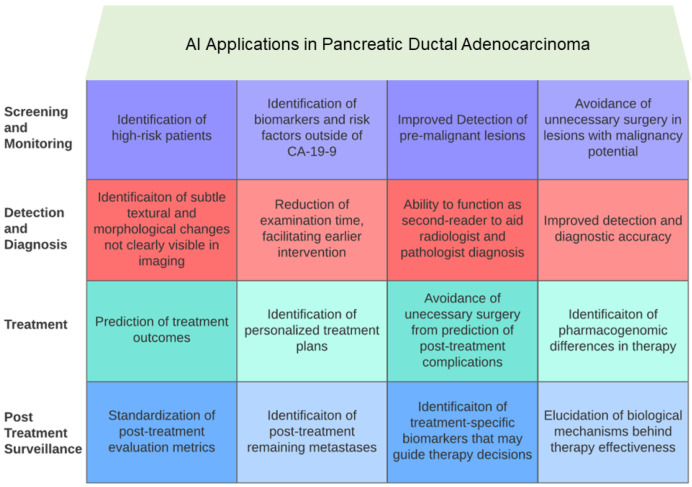
A high-level overview of different areas of impact of AI in PDAC is presented. Each area of impact is reflected in the studies presented in Section 2, Section 3, Section 4, Section 5 and Section 6.

**Figure 2 cancers-16-02240-f002:**
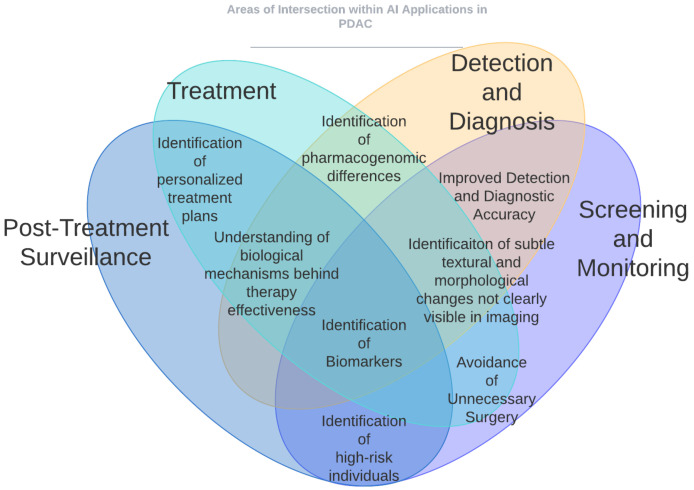
AI applications that intersect different aspects of patient care for PDAC. Identification of personalized treatment plans intersects Section 5 and Section 6. Identification of subtle textural and morphological changes not clearly visible in imaging intersects Section 2, Section 3, Section 4 and Section 5. Avoidance of unnecessary surgery intersects Section 2, Section 3, and Section 6. Understanding of biological mechanisms behind therapy effectiveness and identification of biomarkers intersect Section 2, Section 3, Section 4, Section 5 and Section 6.

**Table 1 cancers-16-02240-t001:** Summary of clinical trials involving AI/ML and pancreatic cancer.

Trial ID	Location	Study	Study Start Date	Enrollment
NCT04743479	Shanghai, China	ESPRIT-AI	20 December 2020	5000
NCT03452774	New York, NY, USA	SYNERGY-AI	1 January 2018	50,000
NCT06055010	Utrecht, The Netherlands	IMPACT	1 January 2014	5000
NCT06320717	Buffalo, NY, USA	AI Derived Biomarker to Select Neoadjuvant Treatment for Borderline Resectable Pancreatic Ductal Adenocarcinoma	2 January 2024	100
NCT04899739	Strasbourg, France	EchoSurg	5 December 2021	45
NCT06256705	Clichy, France	OPERANDI-NET	25 March 2024	80
NCT05729737	Beijing, China	Radiographic Response to Chemotherapy in Unresected Localized Pancreatic Cancer	1 January 2022	100

**Table 2 cancers-16-02240-t002:** Various AI/ML models proposed for PDAC risk prediction or screening.

Category	Ref	Dataset	AI/ML Model	Metrics
Classical ML	[36]	NHIS features	ANN	AUC = 0.85
	[37]	EHR	Logistic Regression	AUC = 0.71
NLP	[38]	Medical Textbooks, Patient Records	Dependency Parser	Sens = 0.99
	[39]	Pathology Reports	Encoder/Decoder + Autoregressive Transformer	F-1 = 0.89
	[40]	ICD-8 + ICD-10	Transformer	AUC = 0.88
CV	[41]	CT Scans	3-D U-Net	DSC = 0.87
	[42]	CT Scans	Naive Bayes	Acc = 0.86

Abbreviations used: ML—Machine Learning, NLP—Natural Language Processing, CV—Computer Vision, ANN—Artificial Neural Network, Sens—Sensitivity, EHR—Electronic Health Record, AUC—Area Under the Curve, NHIS—National Health Interview Survey. Acc—Accuracy, ICD—International Classification of Disease, DSC—Dice Similarity Coefficient.

**Table 3 cancers-16-02240-t003:** Various AI/ML models proposed for PDAC cysts/lesion monitoring.

Category	Ref	Dataset	AI/ML Model	Metrics
Monitoring cysts and lesions	[46]	EUS Confocal Microscopy	Mask-R-CNN + VGGNet	Acc = 0.74
	[45]	CT Scans	LeNet	Acc = 0.89
	[47]	Clinical + Pathological Features	SVM	Acc = 0.77

**Table 4 cancers-16-02240-t004:** Various AI/ML models proposed for PDAC detection/diagnosis.

Category	Ref	Dataset	AI/ML Model	Metrics
Detection	[50]	CT	SVM	Acc = 0.922
	[51]	CEUS	ResNet-50	AUC = 0.953
	[52]	CT	CNNs	AUC = 0.986
	[53]	CT	3D TransUNet	Sens = 0.91
	[54]	EUS	EfficientNetV2-L	Sens = 0.96
	[55]	CECT	3D U-Net	Sens = 0.99
	[56]	CT	ResNet9	AUC = 0.95
Diagnosis	[57]	CECT	SVM	Acc = 0.86
	[58]	CT	VGG16-XGBoost	Acc = 0.97
	[59]	CECT	LASSO Regression	AUC = 0.75
	[60]	CT	CNNs	Acc = 0.867
	[61]	H&E Slides	Bayesian DenseNet-201	Acc = 0.856

Abbreviations used: CECT—Contrast-Enhanced Computed Tomography, SVM—Support Vector Machine, LASSO—Least Absolute Shrinkage and Selection Operation, H&E—Hematoxylin and Eosin, CEUS—Contrast-Enhanced Ultrasound, EUS—Endoscopic Ultrasound.

**Table 5 cancers-16-02240-t005:** Various AI/ML models proposed for PDAC treatment outcome prediction and personalized treatment stratification.

Category	Ref	Dataset	Model	Metrics
Patient Treatment Stratification	[75]	MRI	CNN	AUC = 0.85
	[76]	CT	CNN	Acc = 0.87
Treatment Outcome Prediction	[76]	CT	CNN	Acc = 0.87
	[77]	CT	CNN	AUC = 0.785
	[78]	RNA-seq	VAE/XGBoost	AUC = 0.74
	[79]	IHC	XGBoost	Sens = 0.92
	[73]	H&E	Hover-Net	Hazard Ratio = 2.94

Abbreviations used: IHC—Immunohistochemistry, CNN—Convolutional Neural Network.

**Table 6 cancers-16-02240-t006:** Various AI/ML models proposed for PDAC post-treatment surveillance.

Category	Ref	Dataset	AI/ML Model	Metrics
Post-Treatment Surveillance	[85]	Light Microscopy Images	U-Net	F-1 = 0.80
	[86]	H&E Slides	DenseNet161	F-1 > 0.86
	[87]	H&E	Hover-Net	C-Index = 0.69

C-index—concordance index.

## Data Availability

No new data was created or gathered in this study.

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
