# Peer review of "Pancreatic Ductal Adenocarcinoma (PDAC): A Review of Recent Advancements Enabled by Artificial Intelligence"

_cancers, 2024, doi:10.3390/cancers16122240_

Round 1
Reviewer 1 Report
Comments and Suggestions for Authors
The manuscript titled,” The multifaceted role of miR-21 in Pancreatic Ductal Adenocarcinoma” has several shortcomings as follows:
1. The manuscript title is extremely vague.
2. The manuscript lacks graphical or tabular representation which are the most common form of representation of these datasets.
3. Biggest red flag: How did the authors pick up papers on THIS topic of PDAC treatment is totally missing. The algorithm used, datasets scanned, threshold set should have been clearly elaborated.
4. In the present form this paper is superficial and will not any new information for the readers
Author Response
Reviewer #1:
- The manuscript titled,” The multifaceted role of miR-21 in Pancreatic Ductal Adenocarcinoma” has several shortcomings as follows:
Our response: We thank the reviewer for their feedback, but we want to mention that our review paper's title is “Pancreatic Ductal Adenocarcinoma (PDAC): A Review of Recent Advancements Enabled by Artificial Intelligence”.
- The manuscript title is extremely vague
Our response: We thank the reviewer for their feedback. We believe that the title of our review paper correctly highlights the main topic of our paper: a thorough overview of important and high-impact studies that improve the standard of PDAC care enabled by artificial intelligence and machine learning. The contents of our paper reflect these different studies section-wise in a concise manner.
- The manuscript lacks graphical or tabular representation which are the most common form of representation of these datasets.
Our response: We thank the reviewer for their feedback. Our manuscript currently has six tables and two graphical visualizations. The first table summarizes the ongoing clinical trials involving pancreatic cancer and artificial intelligence, and the rest five summarize the studies described in each section. These tables begin on lines 201, 218,281,333, 383, and 431 of our latex document.
- Biggest red flag: How did the authors pick up papers on THIS topic of PDAC treatment is totally missing. The algorithm used, datasets scanned, threshold set should have been clearly elaborated.
Our response:
We thank the reviewer for their feedback. To address this comment, we can refer to the abstract on line 122. Our document states that, “This review article attempts to consolidate the literature from the past five years to identify high-impact, novel, and meaningful studies focusing on their transformative potential in PDAC management. Our analysis spans a broad spectrum of applications, including but not limited to patient risk stratification, early detection, and prediction of treatment outcomes, thereby highlighting AI's potential role in enhancing the quality and precision of PDAC care. By categorizing the literature into discrete sections reflective of a patient's journey from screening and diagnosis through treatment and survivorship, this review offers a comprehensive examination of AI-driven methodologies in addressing the multifaceted challenges of PDAC. Each study is summarized by explaining the dataset, ML model, evaluation metrics, and impact the study has on improving PDAC-related outcomes. We also discuss prevailing obstacles and limitations inherent in the application of AI within the PDAC context, offering insightful perspectives on potential future directions and innovations.”
- In the present form this paper is superficial and will not any new information for the readers
Our response: We thank the reviewer for their feedback. To address this comment, we can refer to Section 1: Introduction (line 199) in our introduction paragraph to understand how our article differs from existing research that has been done.
Our review paper states that, “The purpose of our study is to consolidate current literature highlighting the use of AI to aid in the screening, diagnosis, and treatment of patients with PDAC. Although other studies exist that have summarized applications of AI in pancreatic cancer, this review differs by aggregating this information according to a patient's chronological timeline and concurrently providing a detailed overview of how each AI/ML application aims to aid in the process [30-33]. Just as a patient first receives screening, gets a diagnosis confirmed, and then gets placed on a treatment plan, the review's sections correspond to a patient's odyssey.”
Additionally, we discuss AI's current limitations in the context of PDAC and offer future solutions regarding molecular data. We have also added effective visualizations encapsulating the major areas in which AI has advanced PDAC screening, diagnosis, treatment, and post-treatment surveillance, allowing readers to capture the main themes within the review paper more easily.
The above-mentioned aspects allow our review paper to add new, meaningful information to AI and cancer researchers, clinicians, and physicians.
Reviewer 2 Report
Comments and Suggestions for Authors
Very interesting and nicely written review on a cutting-edge topic.
I suggest to report, when available, also the agreement between AI and expert endoscopist in the tables and comment this data in the text.
When speaking about the cysts, the authors should comment on the available tool to define the characteristics of these lesions, for example contrast enhancement (in this regard cite the recent SRMA: PMID: 34217751)
Some figures or images would improve the manuscript
The authors should add a table reporting the ongoing studies registered in TrialGov in this field
Author Response
Please find the response to reviewer 2 on pages 4-8.

Reviewer 3 Report
Comments and Suggestions for Authors
The manuscript of the review article "Pancreatic Ductal Adenocarcinoma (PDAC): A Review of Recent Advancements Enabled by Artificial Intelligence" is devoted to the important but narrow problem of improving the diagnosis of pancreatic cancer through artificial intelligence. This work will interest specialists, but it is unlikely to receive wide attention from the scientific community.
The review highlights the problems of the practical application of various techniques of artificial intelligence and machine learning for predicting the risk of complications of the disease and its treatment. Despite the complexity and innovativeness of the issues, the article is written in quite understandable language, and the content of the work is clearly stated.
To improve the quality of work, I recommend adding 1-2 charts; additional data visualization would make this review more attractive.
I support the publication of the manuscript without significant comments.
Author Response
Please find responses to reviewer 3 on pages 8-12. Thank you.

Round 2
Reviewer 1 Report
Comments and Suggestions for Authors
Unfortunately, I donot see any improvement in this revised version of the manuscript.
Reviewer 2 Report
Comments and Suggestions for Authors
The revised version of the paper is OK. Thank you!